

# *CD36* identified by weighted gene co-expression network analysis as a hub candidate gene in lupus nephritis

Huiying Yang and Hua Li

Department of Nephrology, Sir Run Run Shaw Hospital, Zhejiang University School of Medicine, Hangzhou, Zhejiang Province, China

## ABSTRACT

Lupus nephritis (LN) is a severe manifestation of systemic lupus erythematosus (SLE), which often progresses to end-stage renal disease (ESRD) and ultimately leads to death. At present, there are no definitive therapies towards LN, so that illuminating the molecular mechanism behind the disease has become an urgent task for researchers. Bioinformatics has become a widely utilized method for exploring genes related to disease. This study set out to conduct weighted gene co-expression network analysis (WGCNA) and screen the hub gene of LN. We performed WGCNA on the microarray expression profile dataset of GSE104948 from the Gene Expression Omnibus (GEO) database with 18 normal and 21 LN samples of glomerulus. A total of 5,942 genes were divided into 5 co-expression modules, one of which was significantly correlated to LN. Gene Ontology (GO) and Kyoto Encyclopedia of Genes and Genomes (KEGG) enrichment analyses were conducted on the LN-related module, and the module was proved to be associated mainly with the activation of inflammation, immune response, cytokines, and immune cells. Genes in the most significant GO terms were extracted for sub-networks of WGNCA. We evaluated the centrality of genes in the sub-networks by Maximal Clique Centrality (MCC) method and *CD36* was ultimately screened out as a hub candidate gene of the pathogenesis of LN. The result was verified by its differentially expressed level between normal and LN in GSE104948 and the other three multi-microarray datasets of GEO. Moreover, we further demonstrated that the expression level of *CD36* is related to the WHO Lupus Nephritis Class of LN patients with the help of Nephroseq database. The current study proposed *CD36* as a vital candidate gene in LN for the first time and *CD36* may perform as a brand-new biomarker or therapeutic target of LN in the future.

## INTRODUCTION

Systemic lupus erythematosus (SLE) is a chronic, systemic autoimmune disease characterized by autoantibody production, complement activation and immune complex deposition. The incidence of SLE ranges from 0.03‰ to 2.32‰ person-years worldwide (*Rees et al., 2017*).

Lupus nephritis (LN) is one of the most frequent and severe organ manifestations in patients with SLE, the hallmark of which is often glomerulonephritis. Approximately

Corresponding author
Hua Li, h_li@zju.edu.cn

50% of SLE patients develop clinically evident renal disease, up to 11% of whom develop end-stage renal disease (ESRD) at 5 years (*Tektonidou, Dasgupta & Ward, 2016*; *Almaani, Meara & Rovin, 2017*). LN is an important cause of ESRD and mortality. The initial and subsequent therapy of LN mainly consists of immunosuppressants and glucocorticoids, which means there are little efficient and specific therapies. Thus, it is a pressing task to clarify molecular mechanisms involved in LN.

LN is characterized by its complicated physiopathologic mechanism. In LN patients, the formation of immune complexes in different glomerular compartments, the activation of innate immune signal pathways, the infiltration of immune cells and proinflammatory mediators can harm glomerular cells through various approaches (*Devarapu et al., 2017*). Although many studies have determined certain pathological mechanisms of LN, the pathogenesis is still far from clear.

With the development of gene microarray and high-throughput next-generation sequencing, bioinformatics analysis of gene expression profiling has been broadly applied to explore the mechanism underlying diseases and potential diagnostic biomarkers or treatment targets. Among diverse means aiming to investigate altered molecular elements based on comparison between groups of different states, weighted gene co-expression network analysis (WGCNA) is a powerful tool utilized for describing the correlation patterns among genes and exploring hub genes related to certain traits (*Van Dam et al., 2018*; *Langfelder & Horvath, 2008*). WGCNA constructs a co-expression network between genes, and then, genes are divided into several co-expression modules by clustering techniques. Genes in certain module are deemed to share similar biological function and biological process. At last, after relating modules to clinical traits, modules with high correlation to disease are further analyzed and hub genes of pivotal importance to disease are identified. WGCNA has been broadly used for studying of diseases such as cancer (*Jardim-Perassi et al., 2019*), neuropsychiatric disorder (*Huggett & Stallings, 2019*), chronic disease (*Morrow et al., 2015*) and proved to be quite useful.

However, although researchers have conducted numerous bioinformatics studies about LN and have got many achievements (*Arazi et al., 2019*; *Panousis et al., 2019*), WGCNA has rarely been used for studies of LN (*Sun et al., 2019*). In our study, we constructed a co-expression network of the expression profile of glomerulus tissue by WGCNA and confirmed gene modules related to LN. After systematically analyzing the LN-related co-expression module by series of bioinformatics methods, a hub gene associated with LN was identified and verified. Depending on the potential roles of the hub gene in the pathogenesis of LN, we expect to propose novel clues of the diagnosis and treatment of LN.

## MATERIALS & METHODS

### Expression profile data collection
The overall procedures of our study are illustrated in the flow chart (Fig. 1). The gene expression profile of GSE104948 was selected and obtained from the Gene Expression Omnibus (GEO) database (https://www.ncbi.nlm.nih.gov/geo/). The raw data is available

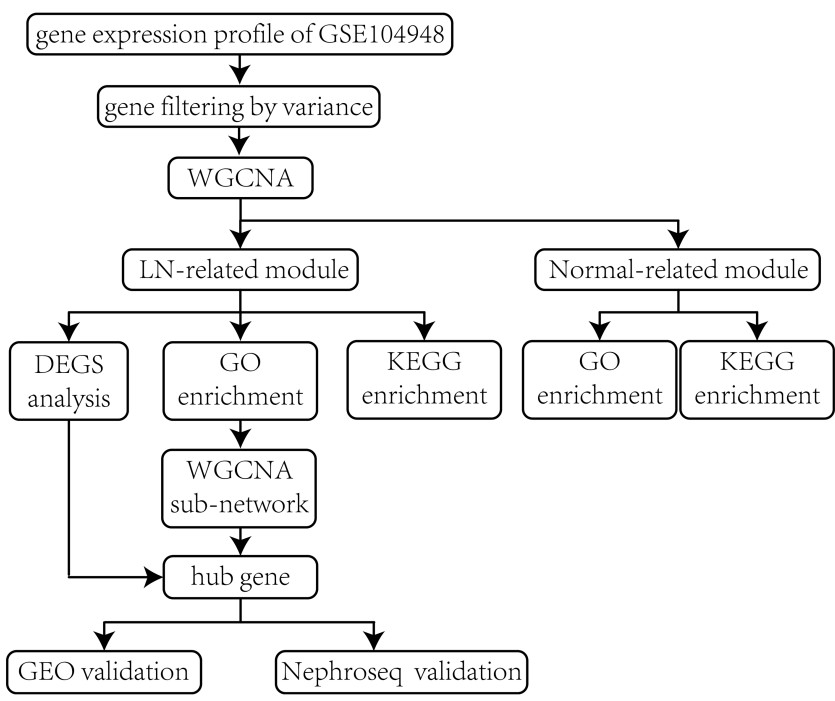

**Figure 1   Flow chart of the whole procedures in this study.** Data processing, analyses, hub gene identification and validation.

in the GEO database. The dataset consists of microarray-based gene expression profiles of 32 LN samples and 18 normal samples of glomerular tissues from SLE patients and living kidney transplant donors respectively. The glomerular tissues were microdissected and verified with glomerular-selective transcripts (*Grayson et al., 2018*). The expression values have already been log2 transformed.

## Data preprocessing

The probe annotation was conducted under the Perl environment with the microarray platform file. Probes matching with multiple genes were removed, and for genes corresponding to multiple probes, the average values of probes were regarded as the expression values of the genes. A total of 11,884 genes were left after the probe annotation.

Since non-varying genes are usually regarded as background noise, we filtered the genes by variance and the top 50% (5,942 genes) with larger variance were chosen for subsequent analyses.

## Weighted co-expression network construction and module division

Before WGCNA, we excluded the outlier samples by sample clustering with the hierarchical clustering method. The sample IDs and details of all samples included in our study are available in Table S1. After that, we applied WGCNA with the expression profile by using the WGCNA package (*Langfelder & Horvath, 2008*) in the R environment (version 3.5.3). Firstly, we calculated Pearson's correlation for all pair-wise genes and constructed a correlation matrix. Secondly, the correlation matrix was transformed to an adjacency

matrix (also known as scale-free network) with an appropriate soft-thresholding value β. A reasonable β value would emphasize strong correlations between genes and penalize weak ones. We calculated the scale-free fit index and mean connectivity of each β value from 1 to 30 respectively, and when the scale-free fit index is up to 0.85, the β value with highest mean connectivity is deemed as the most appropriate one. Then, the adjacency matrix was converted to a topological overlap matrix (TOM) so that the indirect correlations between genes are concerned. Finally, we used average linkage hierarchical clustering according to the TOM-based dissimilarity measure to classify all genes into several co-expression modules with a minimum size of 30 genes, thereby the genes with similar expression patterns were divided into the same module. After defining the first principal component of a given module as eigengene, we calculated Pearson's correlations of the eigengenes, and merged modules whose eigengenes were highly correlated (with Pearson's correlation higher than 0.75) into one module.

To verify the reliability of the division of modules, we plotted an adjacency heatmap of all the 5,942 genes analyzed by WGCNA. Besides, we completed a cluster analysis of module eigengenes and plotted an adjacency heatmap to find out the interactions among modules.

## Identification of clinically significant modules

The clinical traits of our samples included normal and LN, we calculated the correlation between modules and traits. Modules of positive correlation with LN were considered as playing roles in the pathogenesis of the disease. On the other hand, genes in modules of positive correlation with normal trait are indispensable for maintaining normal biological functions. Thus, we extracted gene modules of highest correlation with LN and normal for subsequent studies.

Here, we introduced the definition of gene significance (GS) and module membership (MM), which represent the correlation of a given gene with clinical trait and module eigengene respectively. Genes in clinical-related modules should have high values and preferable correlations of GS and MM.

## GO and KEGG pathway enrichment analyses

To explore the involved signal pathways and biological characteristics of genes in clinical-related modules, we conducted Gene Ontology (GO) enrichment analysis and Kyoto Encyclopedia of Genes and Genomes (KEGG) pathway enrichment analysis and visualized the top 10 significant terms respectively with the clusterProfiler R package (*Yu et al., 2012*). For both of GO and KEGG, enrichment terms arrived the cut-off criterion of *p*-value <0.01 and Benjamin-Hochberg adjusted *p*-value <0.01 were considered as significant ones.

## Differentially expressed genes analysis

To investigate the difference of the expression profiles between normal and LN samples of genes in clinical-related modules, we applied differentially expressed genes (DEGs) analysis based on Empirical Bayes test with the limma R package (*Ritchie et al., 2015*). The cut-off criterion was set as follow: |log2fold change (logFC)|>1; *p*-value <0.01; false discovery rate (*FDR*) <0.001. The results were visualized with the heatmap R package (*Galili et al., 2018*).

### Identification of hub gene

Hub gene of the LN-related module should have high connectivity with the whole module and LN trait, which may play critical roles in the molecular mechanism of LN. For identifying the hub gene related with LN, we extracted gene clusters that enriched in certain GO terms from the WGCNA network to construct sub-networks after GO enrichment analysis of the LN-related module. Then, we utilized the Cytoscape software and its plug-in cytohubba to seek out the hub gene from sub-networks (*Shannon et al., 2003*). After calculating the Maximal Clique Centrality (MCC) value of each gene, those with high MCC values was regarded as hub genes (*Chin et al., 2014*). The results were exhibited with the Cytoscape software. We then surveyed the GS value, MM value and logFC value of the selected hub gene to validate its reasonability.

### Validation of hub gene with the other GEO datasets

To further verify the differential expression level of the hub gene between normal and LN tissues, we analyzed the logFC value of the hub gene with data from the other three GEO datasets (GSE32591, GSE99339 and GSE113342). The GEO IDs and details of the datasets were given in Table S1.

### Validation of the clinical significance of hub gene by Nephroseq database

To assess the relationship between the expression level of the hub gene and the activity or grade of LN, we visited the Nephroseq database (http://v5.nephroseq.org), which provides unique access to datasets from the Applied Systems Biology Core at the University of Michigan, incorporating clinical data which is often difficult to collect from public sources. We then analyzed the difference of the expression level of hub gene between patients in different WHO Lupus Nephritis Class (*Weening et al., 2004*) based on two datasets (the dataset of Peterson Lupus Glom and the dataset of Berthier Lupus Glom) from the Nephroseq database (details available in Document S4). We performed unpaired t test for comparisons between groups and set the criterion of two-tailed value of $p <0.05$ as statistically significant.

## RESULTS

### Data preprocessing

After data preprocessing, 5,942 genes were selected for subsequent analyses. Sample clustering excluded the outlier samples and a total of 18 normal samples and 21 LN samples were left. The final result of sample clustering is shown in Fig. 2A, revealing satisfactory intra-group consistency and distinct difference between groups. The GEO IDs and details about the source of the samples are available in Table S1.

### Weighted co-expression network construction and module division

In the current study, taking both scale-free fit index and mean connectivity as reference, the soft-thresholding was determined as 10 (Figs. 2B and 2C). Accordingly, the correlation matrix was transformed to an adjacency matrix and then converted to a topological overlap matrix. Based on average linkage hierarchical clustering and module merging, the genes

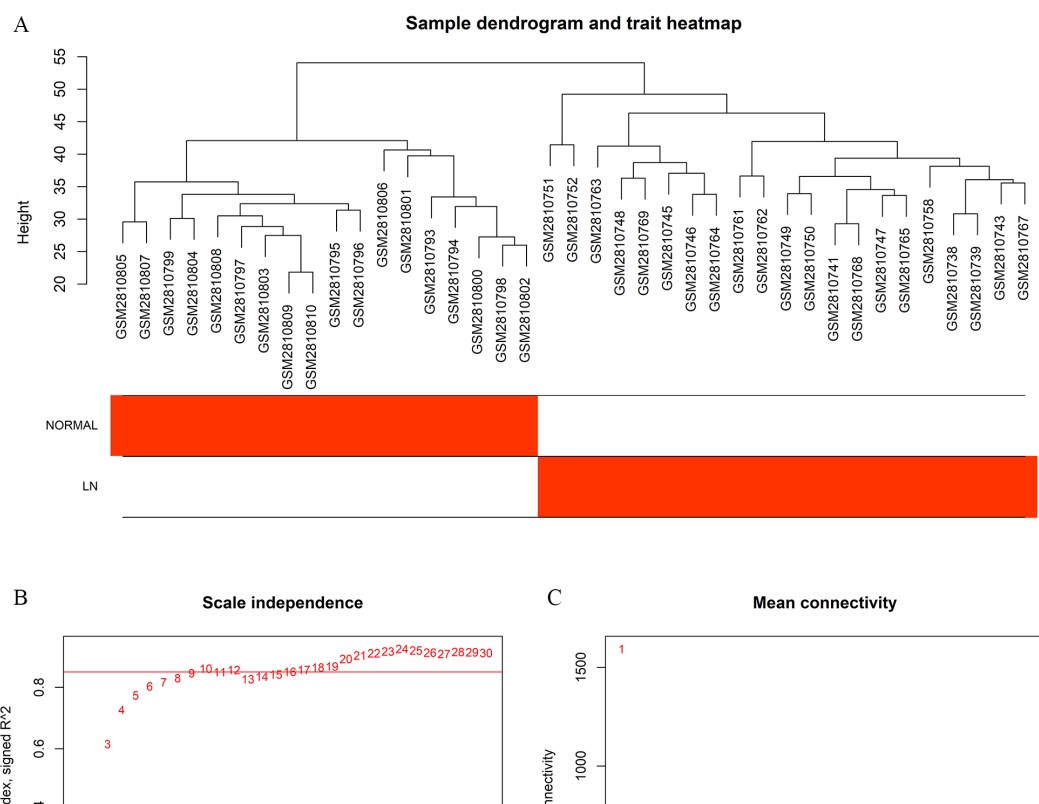

**Figure 2  Sample cluster dendrogram and soft-thresholding values (β) estimation.** (A) Sample cluster dendrogram and clinical trait heatmap of 18 normal samples and 21 LN samples based on their expression profile. (B) Analysis of scale-free fit index of each β value from 1 to 30. (C) Analysis of mean connectivity of each β value from 1 to 30. $\beta = 10$ was chosen for subsequent analyses as it has the biggest mean connectivity when the scale-free fit index is up to 0.85.

were divided into six modules and were displayed with different colors (Fig. 3A), including the black, blue, brown, magenta, pink, and grey modules, containing 220, 2,881, 777, 465, 770, and 829 genes, respectively. Genes in the grey module were those couldn't be divided into any co-expression modules.

Figure 3B depicts the topological overlap adjacency among all the 5,942 genes analyzed by WGCNA, indicating that most genes have higher correlation with genes in the same module and lower correlation with genes in other modules, which means the division of the modules was accurate. The clustering dendrogram and adjacency heatmap of eigengene are shown in Figs. 3C and 3D, meaning that the five modules were mainly separated into two clusters.

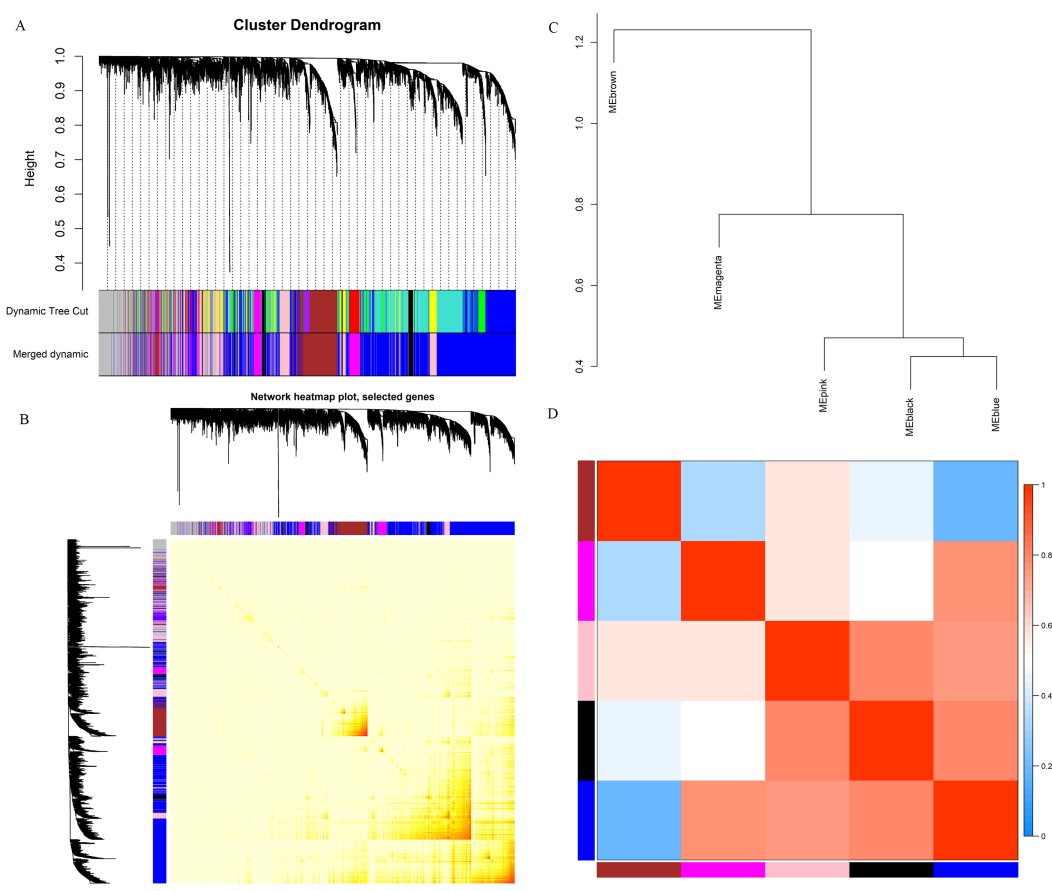

**Figure 3** **Division and validation of co-expression modules.** (A) Dendrogram of all genes divided into 6 modules base on a dissimilarity measure (1-TOM). The modules labeled by color are indicated below the dendrogram. The upper presents the original division by average linkage hierarchical clustering according to the TOM-based dissimilarity measure and the under presents the modules merged according to Pearson's correlation of eigengenes. (B) Adjacency heatmap of the 5,942 genes analyzed by WGCNA. The depth of the red color indicates the correlation between all pair-wise genes. The red color mainly distributes in the diagonal of the heatmap. (C) Clustering dendrogram of eigengenes. (D) Adjacency heatmap of eigengenes. Red represented high adjacency (positive correlation) and blue represented low adjacency (negative correlation).

## Identification of clinically significant modules

We calculated the module-trait correlation coefficients and showed the results in Fig. 4A. The results illuminated that the blue module displayed highest correlation with LN trait ($r = 0.91$, $p = 2e - 15$), while the brown module related best with normal trait ($r = 0.66$, $p = 4e - 06$). The GS and MM value of all member genes of the blue and brown modules were shown in the scatterplots (Fig. 4B and Fig. 4C). The GS and MM value were of high correlation in the two modules (cor $= 0.89$, $p < 1e-200$, and cor $= 0.73$, $p = 3.1e-130$ respectively), suggesting that the genes in the two modules were associate with their module eigengenes and clinical traits synchronously and thus suitable for further analyses and hub gene excavation. We then renamed the two modules as top LN module and top non-LN module respectively.

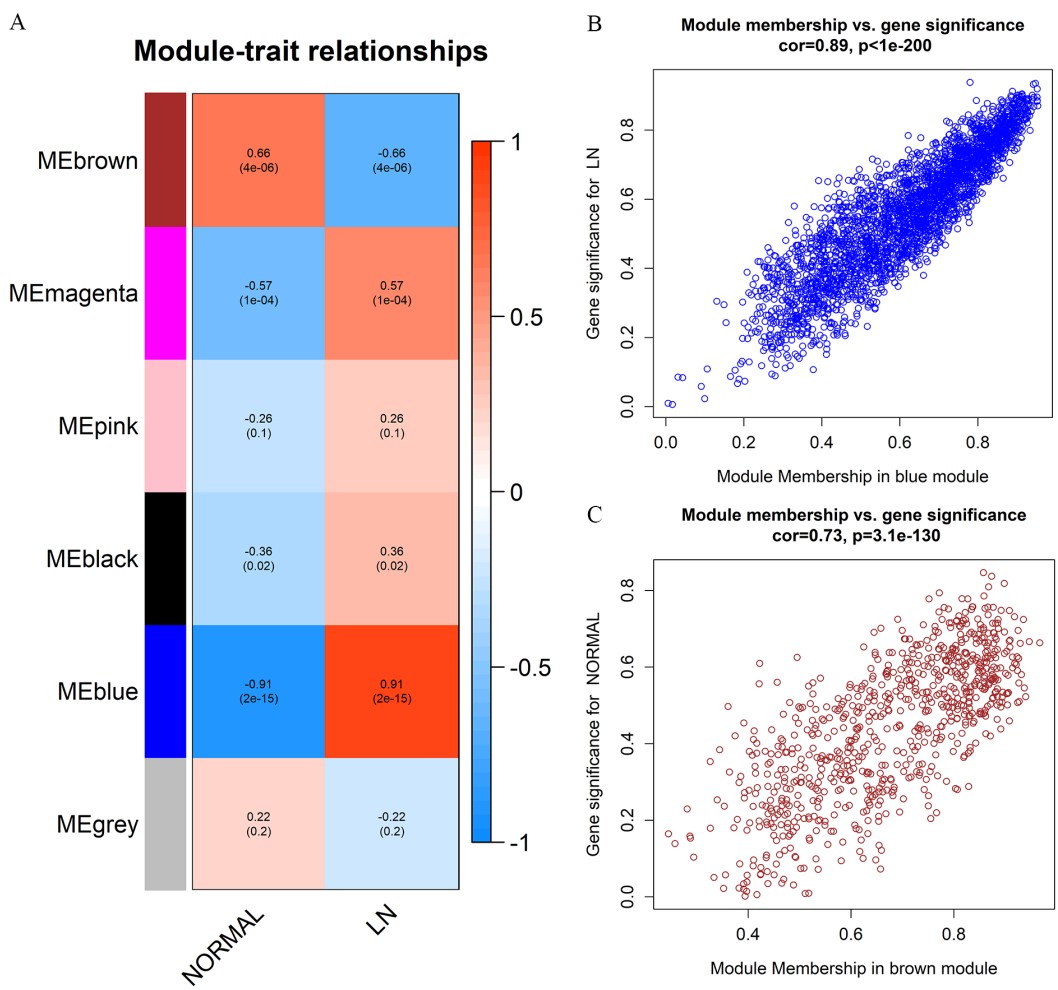

**Figure 4** **Identification and verification of clinical related modules.** (A) Heatmap of module-trait correlations. Each cell depicts the correlation coefficients and *P*-value. The cells are colored by the intensity of correlation according to the color legend (red for positive correlation and blue for negative correlation). The blue module (top LN module) and brown module (top non-LN module) were identified as trait-related modules. (B) Scatter plot for correlation between the Gene significance (GS) and Module Membership (MM) in the top LN module. Correlation coefficients and *P*-value is labeled at the top. (C) Scatter plot for correlation between the GS and MM in the top non-LN module.

## DEGs analysis of trait-related modules

We applied DEGs analysis for the two trait-related modules. For the top LN module, 203 DEGs were screened in LN compare with normal, including 195 up-regulated genes and 8 down-regulated ones. While, the top non-LN module contained 78 DEGs, all of which were down-regulated. The top 30 of up-regulated and down-regulated genes are displayed in Fig. 5, respectively. The logFC value, *p-* value, and *FDR* of DEGs are given in Table S2.

## GO and KEGG Enrichment analyses of trait-related modules

To confirm the biological themes of genes in the trait-related modules and find the underlying biological pathways behind LN, we performed GO and KEGG enrichment

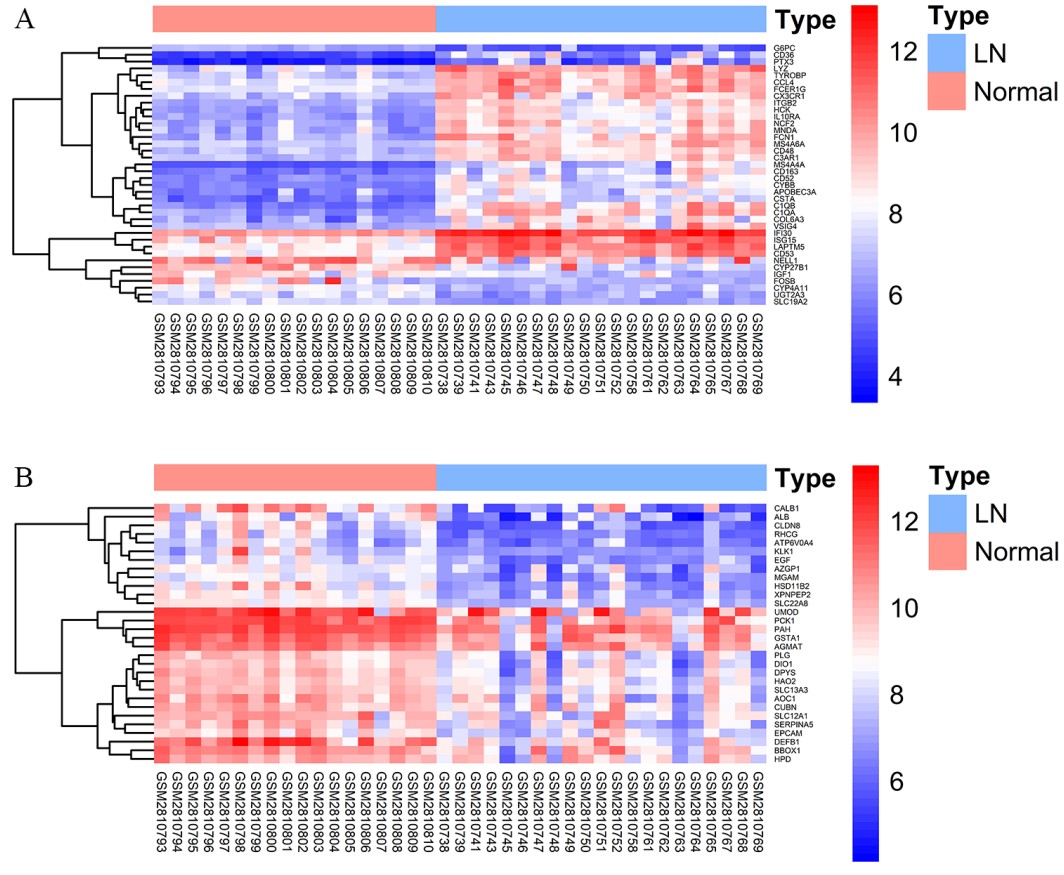

**Figure 5  DEGs analysis of two trait-related modules.** The color of each cell represents the expression level of a gene in a sample (red for high level and blue for low level). Only DEGs with top 30 logFC values are displayed. (A) Heatmap of DEGs in the top LN module. (B) Heatmap of DEGs in the top non-LN module.

analyses towards the top LN and top non-LN modules and the top 10 significant terms of GO and KEGG are exhibited in Figs. 6 and 7 respectively. The complete results of GO and KEGG enrichment analyses were given in Table S3.

For the top LN module, enriched GO-BP terms were mainly about activation of immune response, immune cells, cytokine production, and inflammation (Fig. 6A), such as "neutrophil activation" (gene count = 178, $p = 7.26e-301.52E-20$), "positive regulation of defense response" (gene count = 160, $p = 9.14e-25$), "activation of innate immune response" (gene count = 107, $p = 3.45e-18$). Enriched GO-MF terms were mainly about cytokine and their receptors (Fig. 6B), such as "cytokine binding" (gene count = 39, $p = 2.47e-8$), "cytokine receptor activity" (gene count = 34, $p = 5.3e-7$), "cytokine receptor binding" (gene count = 69, $p = 2.98e-5$). For GO-CC, enriched terms were mainly involved in membrane and endocytic vesicle (Fig. 6C), such as "vesicle lumen" (gene count = 107, $p = 4.13e-15$), "cytoplasmic vesicle lumen" (gene count = 106, $p = 8.99e-15$), "membrane region" (gene count = 95, $p = 2.33e-12$). The results of KEGG enrichment were similar to that of GO-BP, were mainly about immune and inflammation

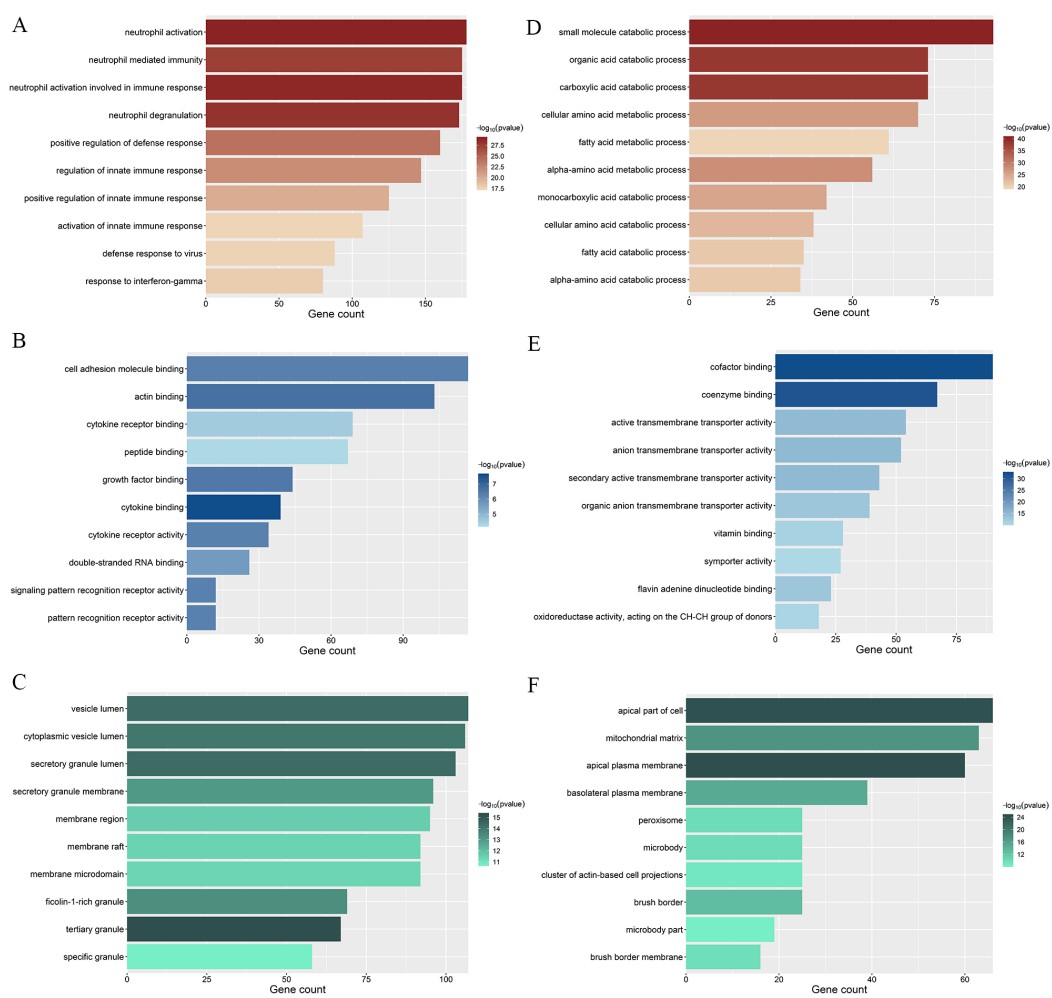

**Figure 6  GO enrichment analyses of two trait-related modules.** The depth of color is corresponded to the enrichment significant of each term and the $x$-axis indicates the enriched gene count. (A) Top 10 significantly enriched GO Biological Process (BP) terms of top LN module. (B) Top 10 significantly enriched GO Molecular Function (MF) terms of top LN module. (C) Top 10 significantly enriched GO Cellular Component (CC) terms of top LN module. (D) Top 10 significantly enriched GO Biological Process (BP) terms of top non-LN module. (E) Top 10 significantly enriched GO Molecular Function (MF) terms of top non-LN module. (F) Top 10 significantly enriched GO Cellular Component (CC) terms of top non-LN module.

(Fig. 7A). The top KEGG terms included "Complement and coagulation cascades" (gene count $= 31$, $p = 6.55\mathrm{e}{-}6$), "Fc gamma R-mediated phagocytosis" (gene count $= 34$, $p = 1.90\mathrm{e}{-}5$), "Th1 and Th2 cell differentiation" (gene count $= 33$, $p = 3.05\mathrm{e}{-}5$). Meanwhile, several well-known pathways involved in LN were also included, such as Th17 cell differentiation.

On the other hand, for the top non-LN module, the GO-BP results were mainly involved in the metabolism process of different kinds of molecule (Fig. 6D), for example, "small molecule catabolic process" (gene count $= 93$, $p = 3.27\mathrm{e}{-}41$), "organic acid catabolic process" (gene count $= 73$, $p = 3.97\mathrm{e}{-}39$), "carboxylic acid catabolic process" (gene

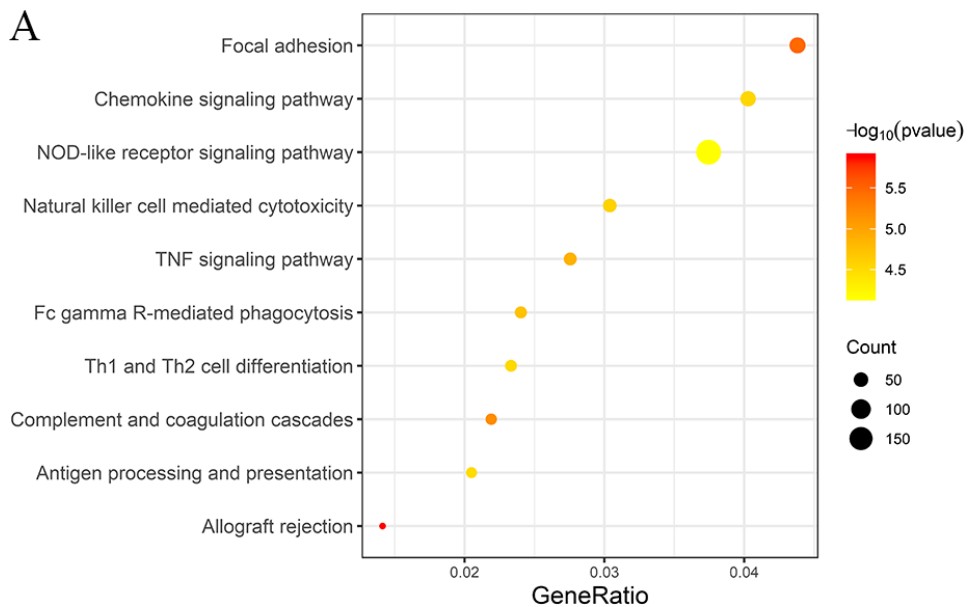

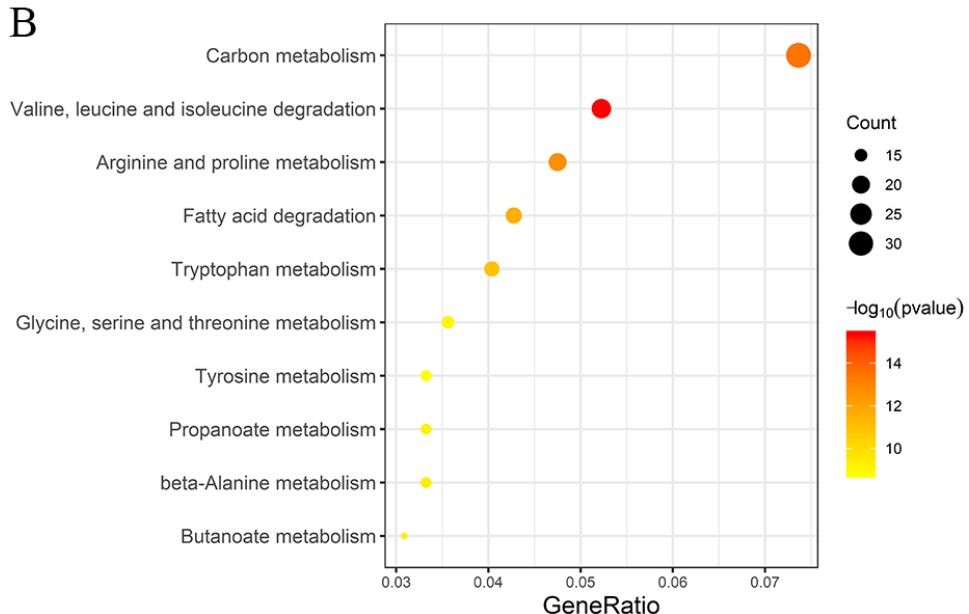

**Figure 7** **KEGG enrichment analyses of two trait-related modules.** The depth of color is corresponded to the enrichment significant of each term and the size of the circle indicates the enriched gene count. (A) Top 10 significantly enriched KEGG terms of top LN module. (B) Top 10 significantly enriched KEGG terms of top non-LN module.

count $= 73$, $p = 3.97e-39$). The top 10 terms of GO-MF and GO-CC are also displayed (Fig. 6E and Fig. 6F). Similarly, the top KEGG terms were mainly about metabolism process (Fig. 7B).

Besides, we found that the magenta module also had high correlation with LN trait. The results of enrichment analyses of magenta module are also given in Table S3.

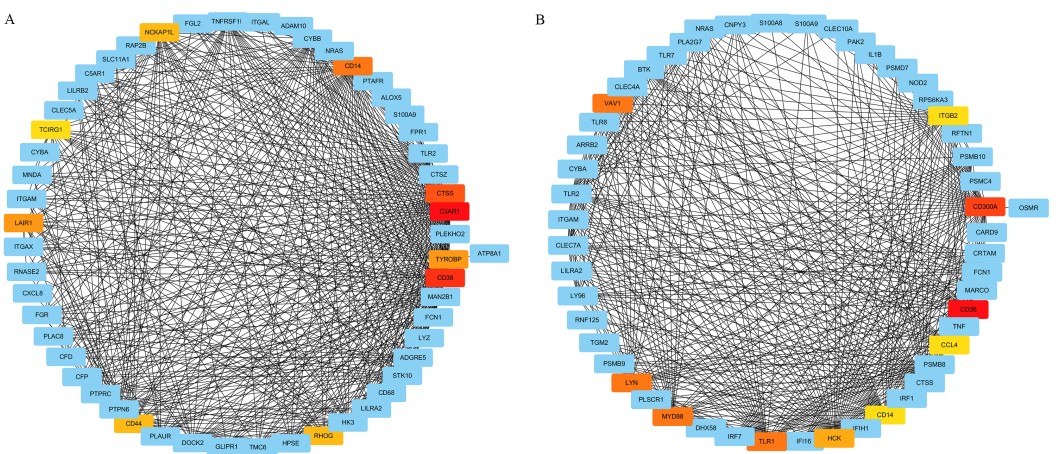

**Figure 8  Sub-networks of WGCNA based extracted based on most significant GO terms.** The nodes represent the genes and the edges represent the weighted correlation. Only co-expression pairs with top 500 weighted correlations are included. The red and yellow nodes represent genes of top 10 MCC values (red for a higher MCC value and yellow for lower). (A) Sub-network of the GO term "neutrophil activation". (B) Sub-network of the GO term "positive regulation of defense response".

## Identification and validation of hub gene

We extracted the genes enriched in two of the top 10 significant GO-BP terms in top LN module, namely "neutrophil activation" and "positive regulation of defense response", and constructed two sub-networks of the weighted co-expression network respectively. Co-expression pairs with top 500 weighted correlations in the sub-networks were selected for hub gene excavation. After importing the gene co-expression pairs and their weighted correlations into Cytoscape, we calculated the MCC value of genes and the most central genes of the sub-network were screened out as shown in Fig. 8. In both sub-networks, *CD36* had the maximum MCC value among all, and was therefore deemed as the hub gene under the pathogenesis of LN.

The GS and MM value of *CD36* were 0.772 and 0.833, revealing *CD36* was significant correlated with the top LN module and LN trait. The DEGs analysis showed that the expression level of *CD36* in LN was abnormally up-regulated compared with that of normal. The logFC of *CD36* was 2.298, which ranked 11th among all.

## Validation of hub gene with the other GEO datasets

For verifying our conclusion in a broader range, we interrogated the GEO database for more datasets about LN. We downloaded the datasets of GSE32591, GSE99339 and GSE113342, and then analyzed the differentially expressed level of *CD36* between LN and normal (Fig. 9). In the four different datasets, the expression level of *CD36* was consistently up-regulated in LN samples, illustrating a satisfactory reliability of the result.

## Validation of hub gene by Nephroseq database

We analyzed the expression level of *CD36* in glomerular tissues under different severity of LN evaluated by WHO Lupus Nephritis Class. The result indicates that *CD36* was

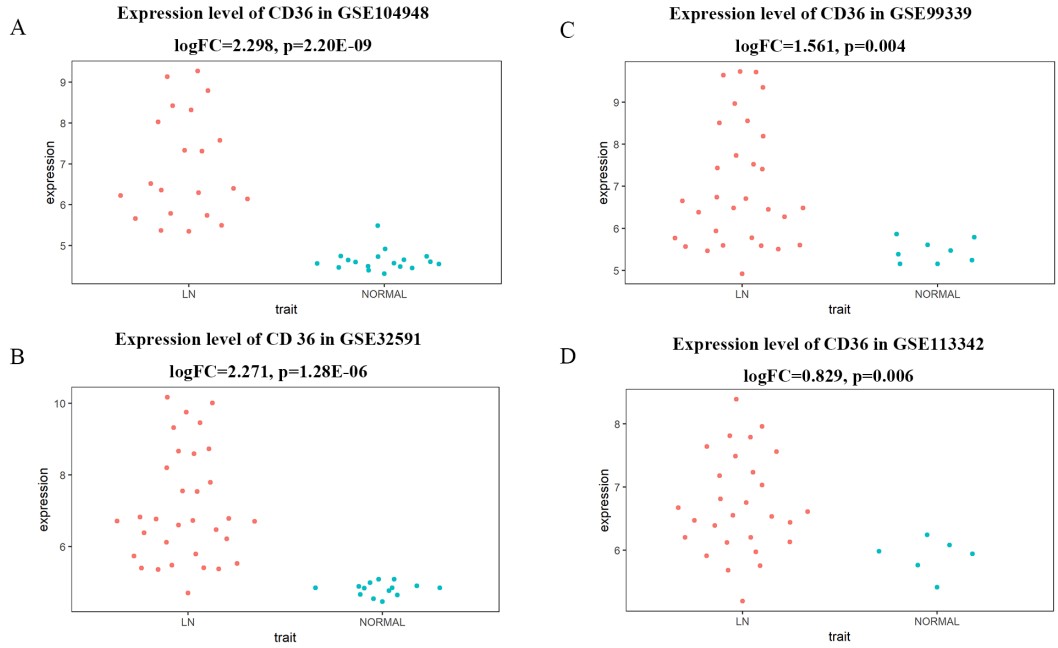

**Figure 9  Differentially expressed level of *CD36* between LN and normal in different GEO datasets.**
(A) Expression level of *CD36* in dataset of GSE104948. (B) Expression level of *CD36* in dataset of GSE32591. (C) Expression level of *CD36* in dataset of GSE99339. (D) Expression level of *CD36* in dataset of GSE113342.

significantly up-regulated with the aggravation of LN (Fig. 10); that is to say, *CD36* plays an important role in the development of LN.

## DISCUSSION

In the current study, we used the expression profile of GSE104948 to screen the hub gene involved in the pathogenesis of LN. We performed WGCNA and divided all genes into 5 co-expression modules. After relating the modules to clinical traits, we concluded that the blue module was of highest correlation with LN and was suitable for hub gene excavating. The brown module had the highest correlation with normal trait, and was also worthy of subsequent analyses. GO and KEGG enrichment illuminated that genes in the top LN module were mostly enriched in biological themes of the activation of inflammation, immune response, cytokine, and immune cells, and the top non-LN module was mainly about the metabolism process of various molecules. What's more, DEGs analysis showed that almost all DEGs in top LN module were abnormally up-regulated, revealing an aberrant activated stage of inflammation and immune response in LN. On the other hand, all DEGs in the top non-LN module were down-regulated, meaning a reduced ability of material metabolism in LN. To achieve the ultimately purpose of finding out the hub gene, we extracted genes enriched in the GO terms of ''neutrophil activation'' and ''positive regulation of defense response'' and constructed sub-networks accordingly. Base on the MCC method, *CD36* with maximum value of MCC in both sub-networks was

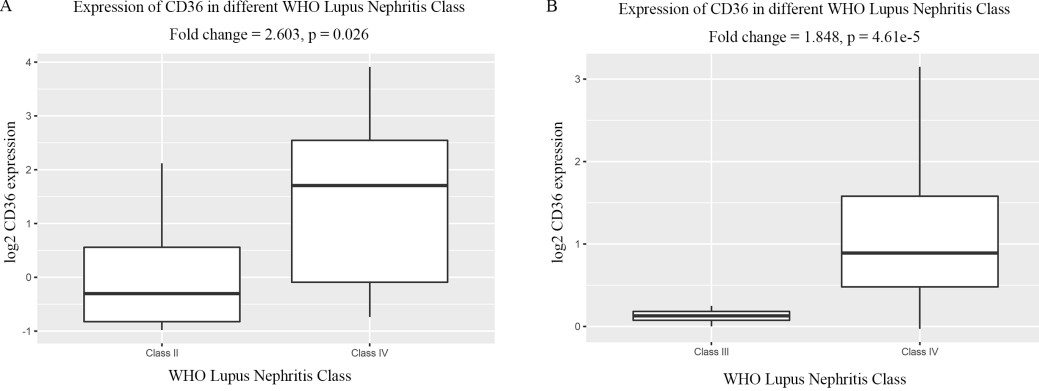

**Figure 10  Differentially expressed level of *CD36* in glomerular tissues of different WHO Lupus Nephritis Class.** (A) Differentially expressed level of *CD36* in class II and class IV respectively (from the dataset of Berthier Lupus Glom). (B) Differentially expressed level of *CD36* in class III and class IV respectively (from the dataset of Peterson Lupus Glom).

regarded as the hub gene behind the pathogenesis of LN. We investigated the GS, MM and logFC of *CD36* and validated its importance. We interrogated the GEO database and got more datasets of LN, the obvious overexpression of *CD36* was therefore further verified. Moreover, the association between *CD36* and WHO Lupus Nephritis Class showed directly that the expression level of *CD36* gradually up-regulates along with the development of LN evaluated by WHO Lupus Nephritis Class, providing strong evidence that the abnormal over-expression of *CD36* is an important element in the pathogenesis of LN.

The *CD36* gene is located on band q11.2 of chromosome 7 (*Fernandez-Ruiz et al., 1993*). The protein encoded by *CD36* is a kind of transmembrane protein (also known as scavenger receptor B2) expresses on the surface of various kinds of cells. In the glomerulus of kidney, *CD36* expresses in podocyte (*Hua et al., 2015*), mesangial cells (*Ruan et al., 1999*) and interstitial macrophages (*Kennedy et al., 2013*). Meanwhile, *CD36* expresses in immunity correlating cells such as monocytes and macrophages (*Collot-Teixeira et al., 2007*). With multiple ligands, *CD36* involves in complex biological process such as lipid homeostasis, immune response and cell apoptosis.

There are no relative reports concerning the direct relationship between *CD36* and LN. Our results have shown that *CD36* mainly participates in the function of neutrophil, such as "neutrophil activation", "neutrophil activation involved in immune response", "neutrophil degranulation", "neutrophil mediated immunity", as well as the activation of immune response, such as "positive regulation of defense response", "positive regulation of innate immune response", "innate immune response-activating signal transduction", "positive regulation of immune effector process". Base on the results, we conclude that *CD36* performs important functions in the pathogenesis and development of LN through affecting the function of neutrophil and innate immune response. Studies have proved that deposited immune complexes (IC) could activate complement and attract neutrophils and potentiate their responses, which will lead to intense glomerulonephritis, release protease and Reactive Oxygen Species (ROS), and give rise to kidney involvement of SLE

(*Tsuboi et al., 2008*). Besides, IC-induced activation of neutrophils can lead to the formation of neutrophil extracellular traps (NETs), which can be pathogenic and promote the release of type I interferon (*Garcia-Romo et al., 2011*). *CD36* may candidate in the pathogenesis of LN by the above-mentioned pathways.

Numerous studies have shown that *CD36* participates in the pathogenesis of several kinds of chronic kidney disease (CKD).

It has long been known that chronic inflammation is an important segment of the progression of CKD. Studies have proved that the ligands signal via *CD36* to promote inflammatory response and the recruitment and activation of macrophage in the glomerulus (*Kennedy et al., 2013*). A report of LN showed that renal macrophage is associated with onset of nephritis and indicates poor prognosis (*Bethunaickan et al., 2011*). Meanwhile, oxidant stress plays a critical role in glomerular dysfunction. Along with chronic inflammation, *CD36* may facilitate the development of oxidant stress in LN (*Hua et al., 2015*; *Kennedy et al., 2013*; *Aliou et al., 2016*).

Podocyte is most susceptible to injury among the component of the glomerulus and its injury leads to glomerular dysfunction in various renal diseases including LN. Here, we determined that podocyte functional markers were down-ragulated in LN glomerular tissue, including *WT1* (logFC = −0.273, $p$ = 0.009) and *NPHS1* (logFC = −0.306, $p$ = 0.034), indicating the fact of podocyte injury. It is reported that in primary nephrotic syndrome mouse, the overexpression of *CD36* in the podocyte promotes its apoptosis (*Yang et al., 2018*). There is probably similar pathogenesis in the progression of LN.

Ectopic lipid deposition in kidney may cause lipotoxicity and further affect the function of the kidney (*Lin et al., 2019*). *CD36* is a multifunctional protein function as a key molecule in the uptake of long-chain fatty acids, which is the main component of fatty acids uptake system in the kidney and plays a critical rule in the development of CKD (*Gai et al., 2019*). The expression level of *CD36* is higher in kidney with acute or chronic damage, and lipid disorders will stimulate the up-regulation of *CD36*. Furthermore, *CD36* can promote the uptake of lipid from plasma to tissue (*Hua et al., 2015*; *Lin et al., 2019*; *Nosadini & Tonolo, 2011*; *Yang et al., 2017*). Among our results, the top 10 GO terms of non-LN module includes two terms about lipid metabolism: fatty acid catabolic process ($p$ = 8.03e−22) and fatty acid metabolic process ($p$ = 2.62e−20), implying that lipid disorders exist in the glomerular tissue, in which *CD36* may take part.

Regrettably, as most reports on the relationship between *CD36* and kidney diseases are about the disorders of kidney in metabolic diseases, there is no study about the role *CD36* plays in immune response, which is worthy of further study in LN.

## CONCLUSIONS

In conclusion, through WGCNA and a series of comprehensive bioinformatics analyses, *CD36* was confirmed for the first time as a hub gene in the pathogenesis of LN. *CD36* is likely to become a new biomarker or therapeutic target of LN. Our work might provide a new insight for exploring the molecular mechanisms of LN.

### Funding

The authors received no funding for this work.

### Competing Interests

The authors declare there are no competing interests.

### Author Contributions

- Huiying Yang conceived and designed the experiments, performed the experiments, analyzed the data, contributed reagents/materials/analysis tools, prepared figures and/or tables, authored or reviewed drafts of the paper.
- Hua Li authored or reviewed drafts of the paper, approved the final draft.

### Data Availability

Data is available at NCBI GEO: GSE104948.

### Supplemental Information

Supplemental information for this article can be found online at http://dx.doi.org/10.7717/peerj.7722#supplemental-information.

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
