# Peer review of "CD36 identified by weighted gene co-expression network analysis as a hub candidate gene in lupus nephritis"

_PeerJ, doi:10.7717/peerj.7722_

## Round 0.1 · original submission · Major Revisions

Please carefully address all the critiques of both reviewers and revise manuscript accordingly.

Reviewer 1 ·

Basic reporting

Yang et al report the results of a gene co-expression network analysis in the context of a manifestation of systemic lupus erythematosus, lupus nephritis, to identify hub genes that could represent new biomarkers of disease and/or potential therapeutic targets. The authors analyzed previously published gene expression microarray data and identified CD36 as hub gene associated to the condition and then try to validate the results using g additional gene expression data from four microarray datasets.
The English language is globally understandable, despite the use of a few terms seems unusual (i.e. excavation instead of Identification at line 204, “to distinguish the biological themes of gene” line 170).
The introduction provides a clear context for the study. References should be added at line 69, when authors refer that WGCNA has been rarely used for studying LN. The structure of the paper is clear. Some figures should be improved for clarity. Figure 3A showed the dendrogram with the results of two different clustering methods. The authors should specify in the text that they chose the merged dynamic method and justify their choice. Figure 3B show a different heatmap dendrogram compared to the modelled gene network: this can lead to confusion. Eigengene adjacency map provided in Figure 3 should be improved by adding at least the labels indicating module color. The authors should consider to provide also a clustering representation together with the heatmap to improve the interpretation of the figure.
Finally the rawdata are accessible by GEO, so that readers can identify original data.

Experimental design

The research question is well defined and relevant and the work has the potential to make the field moving forward.
Concerning the overall design of the study, a major question is whether the results are linked to the true pathogenesis of the disease or to the fact that in LN the tissues are infiltrated with immune cells, and the different gene expression profiles between control tissues and LN are simply due to cell type composition. The authors should comment on this point, eventually providing additional information on the pathology of the studied tissues.
Concerning the co-expression analysis, I wonder why the authors restricted their discovery analysis using WGCNA to only one gene expression dataset, when three out of the four datasets used for the validation (Figure 9) are based on the same or very similar microarray platform (all of them being Affymetrix arrays). The same analysis performed on one microarray should be performed on the pooled data and validated by the fourth dataset based on Nanostring technology. This approach should be tested to provide results that are expected to be statistically more robust and biologically more meaningful. Finally, the authors should state why they discarded other modules still significantly associated to LN in their subsequent analysis (lightyellow and darkgrey).

Validity of the findings

While the authors provide some evidence that CD36 could be an important gene in LN, the limits of the design do not allow a conclusion on the validity of the results.

Additional comments

No other major comments

·

Basic reporting

General comment
Reusing publicly available genomic data for different bioinformatics approaches can help reveal novel candidate genes responsible for disease. The present work is interesting and the authors have managed to find a differentially expressed gene that could be relevant to Lupus nephritis. However, important details are missing from the manuscript and the relevance of the proposed hub gene has not been confirmed in any way. I think that the manuscript should be thoroughly revised to be suitable for publication.

Experimental design

Specific comments:
1. In Material and Methods – Expression profile data collection: I checked the GEO accession number provided by the authors and I was not able to figure out which samples were included in the study. Please give more details on the study group including preferably a supplementary table with the GEO IDs and associated information. Also, provide more details on the type of controls you are using (what’s normal and why did they undergo a biopsy?).
2. In Material and Methods – Weighted co-expression network construction and module division: Please specify whether any variance-related filtering was performed. Also, were lowly variable/non-expressed genes taken into consideration in the WGCNA analysis? If so, you are at a high risk of including genes in your study with weak biological significance. Please detail this part.
3. In Material and Methods – GO and KEGG pathway enrichment analysis: You point out that a nominal p value<0.01 and a BH-corrected p value<0.05 are used for GO and KEGG pathway enrichment analyses (respectively, I guess). These thresholds seem to be arbitrary and one cannot understand why you use a corrected p value in a case and a nominal p value in the other. Please explain your choice or be more consistent and restrictive.
4. In Material and Methods – Differentially expressed gene analysis: Same comment as before. You perform a genome-wide analysis but do not correct for multiple comparison. Additionally, this results in an extremely large heat map that is difficult to read and interpret. I would be more restrictive, at least for data visualization.
5. In Material and Methods – Identification and validation of hub gene: You mention different GEO datasets that you used for the validation of the hub gene. However you do not provide any information (not even accession numbers) nor describe the sample size and characteristics. I propose to provide this information in the manuscript and to complete the abovementioned supplementary table with IDs/associated info corresponding to these confirmatory datasets as well.

Validity of the findings

Specific comments:
6. Results – Data processing: You mention a clustering for sample filtering. Can you provide it as a supplementary figure or at least make it clear which are the final samples from GEO included in the analysis?
7. Results – Identification of clinically significant modules (exclude DEGs from the tittle in this section, you speak about them in the following section and not in this one): I don’t think the authors interpret correctly Figure 3c. In fact, it represents adjacency between eigengenes from different modules and therefore, do not provide any information about the accuracy in the division of the modules (as stated in the last sentence of this section), but the relationship among them. Moreover, you don’t interpret further such a relationship and I do not think you should mention in Results anything that is not going to be discussed. Please revise both the figure legend and the main text. Please specify what dynamic tree cut and merged dynamic mean in your dendogram (Figure 3A).
8. Results – Identification of clinically significant modules: At the end of the section you mention that your results reveal that the darkgreen/lightgreen modules are suitable for further investigation. I think the most important finding here is the strong positive correlation you find between module membership and trait significance, suggesting that the highest the membership the highest the association with LN or normality. I think this should be highlighted, probably in both Results and Discussion. Additionally, you should rename your main modules (colors do not have any biological meaning; I would replace colors by something like top LN and top non-LN modules). Finally, I think you should remind the readers about the number of genes contained in the further investigated modules.
9. Results – Enrichment analysis and DEG analysis of trait-related modules: The last paragraph of this section is highly speculative and should be rewritten and moved to Discussion.
10. Discussion: I miss more references to previously performed trascriptomic analyses of LN.
11. Discussion: I think there are parts in the Discussion that are too speculative/ overoptimistic in the interpretation of the Results.
12. Discussion: There are a number of genes appearing for the first time in the Discussion. I think being more restrictive in the DEG analysis would give as a result a shorter candidate gene list and a more comprehensive heat map (please revise Figure 7). I think the text would flow better if you mentioned at Results anything you would like to discuss later.

Additional comments

13. I miss a survival analysis that proves the importance of CD36 in LN. From my point of view, overexpression is not a strong enough evidence to prove the centrality of CD36. Could you please try to perform such an analysis by using for example TCGA data (if available)? I think a positive result would substantially increase the impact of this work.
14. Please mention abbreviations at first appearance and revise English.

---

## Round 0.2 · Minor Revisions

Please address remaining concerns of the reviewer #2

Reviewer 1 ·

Basic reporting

I believe the authors have been addressed the major criticisms I have raised.

Experimental design

Clarifications have been made on the design and it is now clear the choice to include only one dataset for the co-expression analysis and to use the others for further validation.

Validity of the findings

The new analysis provided by the authors seems to validate the previous results.

·

Basic reporting

The authors have made substantial changes to the manuscript including the reanalysis of the data after a variation-based filtering and a differential expression analysis according to the LN class in an independent database. Additionally, they have detailed the Methods section as required by the Reviewers. In general, I think that the manuscript has considerably improved and I congratulate the authors for they thorough revision.

Experimental design

The Methods section is more accurate now. A supplemmentary table and document have been added to provide details and IDs on the specific samples used in the analysis.

Validity of the findings

Some of the Figures have been redone, new analyses have been conducted. Discussion has been rewritten to reduce the speculative considerations.

Additional comments

Minor comments:

-I would mention Supplementary Table 1 in Methods for the first time.

-Sorry for suggesting TCGA (I was thinking about a similar database on Lupus but obviously TCGA is not the one). I think Nephroseq was a good choice and certainly, your new analysis strenghtens your previous results. However, you do not comment about this in the Discussion. Moreover, you plot class II and III vs IV separately (Figure 10), and group IV looks different in the two plots although it is supposed to be expression data of the same samples, right? Please, pay attention to this Figure and accordingly discuss in the appropriate section.

---

## Round 0.3 · accepted · Accept

All remaining critiques were adequately addressed and the manuscript was amended accordingly.